# Economic suitability of direct seeded rice across different geographies in India

**Shiladitya Dey**☯, **Kumar Abbhishek**[iD]☯*, **Suman Saraswathibatla, Debabrata Das**

Action for Climate and Environment, Dr. Reddy's Foundation, Hyderabad, Telangana, India

☯ These authors contributed equally to this work.
* abbhishek.k@drreddysfoundation.org

## Abstract

Puddled transplanted rice (PTR) is being replaced by dry direct-seeded rice (dDSR) to address manpower, water, and agricultural costs. The economic suitability of dDSR in different agro-climatic regions limits its widespread adoption. We use plot and household data to estimate the impact of dDSR adoption in four Indian rice-growing states. We first used propensity score matching (PSM) to assess how dDSR adoption affected operation-wise cultivation costs, paddy yield, and net income. The yield effect on DSR adoption was estimated using endogenous switch regression (ESR) to account for observed and unobserved heterogeneity. Both PSM and ESR-based results show that DSR adoption may increase paddy yield in Uttar Pradesh, Andhra Pradesh, and Telangana and decrease it in Madhya Pradesh, but net income from paddy farming increased significantly (Rs5009/acre to 8134 based on different locations) in all four states. Adopting dDSR helps resource-poor Indian farmers reduce paddy production costs and increase income. Therefore, Central and State governments must implement policies and strategies to encourage non-adopters to adopt dDSR.

## Introduction

Rice is essential for global food security, serving as a staple for half of the world's population. [1]. India leads in paddy cultivation area and ranks second in production. India produces 21% of global rice [2]. Global demand for milled rice is projected to increase by 96 million tonnes by 2040 compared to 2015 [3]. Rice production must rise for food security and sustainability. Furthermore, enhanced production must be ecologically sustainable, utilizing less water, labour, and agrochemicals. This can be accomplished by bridging rice output gaps and conserving natural resources like soil and water, particularly in India's Indo-Gangetic Plain and Southern States.

Groundwater is used extensively in conventional paddy cultivation. Puddled Transplanted Rice (PTR) involves physically transplanting paddy into tilled and puddled soil. Evaporation and percolation consume 60–80% more water than required by the crops, raising paddy cultivation costs [4,5]. Consequently, water-efficient, ecological, and cost-effective paddy cultivation necessitates water conservation [6]. Dry-direct seeded rice (dDSR) saves water in paddy production, where seeds are directly sown in the main field and established by pre-monsoon rainfall [6]. The majority of research shows that dDSR reduces paddy water use by 35–57%

**Data availability statement:** All relevant data are within the manuscript and its Supporting Information files.

**Funding:** The author(s) received no specific funding for this work.

**Competing interests:** The authors have declared that no competing interests exist.

compared to PTR, although the total volume of saved water varies greatly with soil type [5–7]. Nevertheless, there is apprehension that water reduction lowers dDSR yield [8,9]. Therefore, it is crucial to determine whether dDSR adoption is economically viable in different geographies with different soil types and properties.

DSR productivity is often lower than PTR due to panicle sterility, poor crop stand, micronutrient deficiencies, higher weed pressure, and root-knot nematodes. Conversely, research also reports that DSR yields 2–3 t ha$^{-1}$ more grain than PTR due to improved panicle number, test weight, and lower sterility percentage [10,11]. Table 1 encapsulates these conflicting data, indicating that DSR may not consistently outperform PTR in analogous rice ecologies. Therefore, it is essential to ascertain the economic and yield viability of DSR across various geographies.

Such performance-based effectiveness of dDSR has been further reported in various Indian agro-climatic zones [16]. In summary, dDSR is a promising technology in rice ecologies; nevertheless, soil type, water management, and climate may constrain its efficacy [17–20]. We observed that most of the studies compared PTR and DSR irrigation water savings and yield using experimental plots with a mention of long-term changes in soil chemical and physical properties, greenhouse gas emissions and global warming potential [21–26]. While extensively examining the various impacts of DSR on environmental and crop performance, studies have notably failed to ascertain whether resource-constrained small and medium farmers may benefit from DSR compared to PTR. Moreover, the suitability of DSR across various agro-climatic situations from an economic standpoint is in question. This study evaluates DSR establishment methods in India's four rice-growing agro-climatic zones. Our plot-level data comes from Uttar Pradesh (UP), Madhya Pradesh (MP), Andhra Pradesh (AP), and Telangana (TS). This study enriches the current literature in three ways. First, it utilises farm-level survey data to evaluate the effects of DSR on rice productivity, operational cultivation costs, and net revenue, addressing the limitations of experimental plot data that may not correctly represent the realities of paddy farmers. [27–30]. Second, endogenous switching regression accounts for observed and unobserved heterogeneity [31]. Third, the study evaluates DSR's economic viability using yield, income, and cultivation costs for four agro-climatic zones. We report that we could not find any such analysis of the DSR's economic impact using survey data from four different agro-climatic conditions.

## Methods

### Propensity score matching method

The decision to adopt the DSR method is a binary choice for the farmer. Farmers will use the technology if the net gain from adopting DSR is greater than not adopting it. The farmers' net income from adopting DSR compared to non-adoption is represented as $K^*$ (Equation (1)). If $K^* > 0$ indicates that the farmer's net gain from adopting DSR is greater than the benefit from not adopting it. However, $K^*$ is not possible to be measured directly. Nonetheless, it

**Table 1. Paddy yield comparison in DSR and PTR.**

| Author(s) | Country | Rice ecology | PTR (t ha$^{-1}$) | DSR (t ha$^{-1}$) |
|---|---|---|---|---|
| Dey et al. (2024) [12] | India | Favorable irrigated | 6.42 | 6.92 |
| Mishra et al. (2017) [13] | India | Favorable irrigated | 7.26 | 7.53 |
| Panneerselvam et al. (2020) [14] | India | Rainfed land | 5.14 | 5.44 |
| Singh et al. (2009a) [15] | India | Rainfed lowland | 6.8 | 6.6 |

can be expressed as a function of quantifiable components inside the latent variable model described below:

$$K^* = \alpha Z_i + \varepsilon_i, \quad K_i = 1[K_i^* > 0] \tag{1}$$

where $K_i$ is a binary variable equal to 1 if the household has adopted the dDSR and 0 if it has not. The coefficient $\alpha$ represents the parameters that need to be measured. $Z_i$ represents the characteristics of households and farms in vector form. The error term $\varepsilon_i$ is assumed to follow a normal distribution. Equation (2) illustrates the probability of implementing the DSR establishment method:

$$\Pr(K_i = 1) = \Pr(K_i^* > 0) = \Pr(\varepsilon_i < \alpha Z_i) = 1 - G(-\alpha Z_i) \tag{2}$$

Here, $G$ represents the cumulative distribution function for $\varepsilon_i$. Regression models such as logit and probit are typically derived based on the assumptions about the functional form of $G$. Various socioeconomic and demographic parameters, yield, and net revenue from production are expected to influence the adoption of DSR technology. In order to establish a connection between the decision to adopt DSR and the possible consequences of adoption, we will examine a risk-neutral production system that aims to maximize net return $\ni$. This system operates in a competitive market for both output and input and utilizes a single-output technology that is quasi-concave in the vector of variable inputs, $H$. This can be expressed as equation (3):

$$max \ni = CP(H, Z) - LH \tag{3}$$

$R$ is the selling price of the paddy, $P$ is the expected production level, $L$ is a common vector for input costs, and $H$ describes the attributes and features of households and farms. The net return from adopting DSR can be represented as a mathematical function of the choice of DSR adoption K, the output price, input variables, and household features, as shown in Equation (4).

$$max \ni = \ni(K, L, R, Z) \tag{4}$$

Equations 4 suggest that the decision to adopt DSR technology, output and input pricing, and household and farm characteristics can affect input demand, net return, and farm productivity.

The PSM method can offer an impartial evaluation of the impact of the treatment (i.e., adoption of DSR), provided that the outcomes are not influenced by the assignment to treatment, given the pre-treatment baseline covariates. The PSM approach primarily quantifies the impact of the treatment on the population that received it, as represented by equation (5).

$$\forall \vert_{K=1} = T(\forall K = 1) = T(V_1 1) - T(V_0 1) \tag{5}$$

Where $\forall$ is the average treatment effect on treated (ATT), $V_1$ indicates the outcome values of the DSR adopters, and $V_0$ is the value of the same variables for non-adopters. However, the study does not estimate $T(V_0 K = 1)$. Instead, the study measures the difference $(\forall^e)$ between $T(V_1 K = 1)$ and $T(V_0 K = 1)$. Hence, $\forall^e$ acts as a potential bias estimator.

The PSM approach can mitigate the sample selection bias when experimental and/or panel data are unavailable [32]. The PSM model uses conditional probability to determine the likelihood of farmers adopting DSR technology depending on their pre-adoption characteristics

[33]. The PSM model relies on the unconfoundedness assumption, also known as the conditional independence assumption. This assumption implies that, after controlling for $E$, DSR technology adoption is random and unrelated to the outcome variables. PSM can be represented by Equation 6.

$$p(E) = \Pr\{K = 1E\} = T\{K \mid E\} \tag{6}$$

Where $K = \{0,1\}$ is the indicator for DSR adoption, and $E$ is the pre-adoption attributes. The conditional distribution of $E$ in given $p(E)$ is the same in both DSR adopters and non-adopters.

The PSM technique does not necessitate making assumptions about the functional form, which involves describing the relationship between outcomes and predictors of outcomes. The primary limitation of the PSM technique is the assumption of unconfoundedness. Even after conditioning, there may still be systemic variations in the outcomes of non-adopters and adopters due to unmeasured baseline variables. However, the PSM technique allows checking the specification to eliminate biases that exceed the average.

Once the propensity score (PS) has been established, the average treatment effect on the treated (ATT) can be calculated using equation 7:

$$\forall\, = T\{(V_1 - V_0 K = 1)\} = T\{T(V_1 - V_0 K = 1), p(E)\} = T\left[T\{(V_1 K = 1), p(E)\}\right]$$
$$- \left[T\{(V_1 K = 0), p(E)\}\right] \tag{7}$$

Various matching strategies can be employed to match adopters with non-adopters who have similar PS. This study utilizes nearest neighbor matching (NNM), kernel-based matching (KBM), and radius matching (RM) to assess the reliability of the results.

## Endogenous switch regression approach

PSM method reduces selection bias caused by observables. However, it does not affect selection bias produced by the unobservable. The PSM method limits conclusions to individuals or farmers whose attributes are present in both the sample and control groups. In this study, we aim to measure the effects of the adoption of DSR over PTR methods on costs, productivity, and income through cross-sectional household data. It is crucial to mention that, like in past research, in this study, the treatment and control groups are not randomly allocated [34]. Smallholder households may self-select into the treatment group by adopting DSR, an example of endogeneity and self-selection. The second concern is estimating the impact of the adoption of DSR through an empirical method. However, in our case, we anticipate that household and farm characteristics may affect DSR's adoption, which further influences outcome variables.

The Endogenous Switch Regression (ESR) model can overcome the abovementioned concerns associated with impact assessment. The ESR model treats the adoption of DSR as a regime shifter. Moreover, ESR considers the observed and unobserved variations among smallholders in the two adoption schemes. Two steps make up the ESR regime. An initial step is a binary choice criterion function or selection equation. In such a situation, the smallholder will evaluate the available resources and management alternatives before deciding whether to adopt DSR. The farmer matches the expected utility of DSR adoption, $D^*_{i,DSR}$, to the expected utility of PTR ($D^*_{i,PTR}$, or Conventional method). Farmers will adopt DSR if $D^*_{i,DSR} > D^*_{i,PTR}$ and will not adopt if $D^*_{i,DSR} < D^*_{i,PTR}$. $D^*_i$ is the adoption dummy variable that is unobservable, but we do observe $D_i$. Initially, we will estimate with probit (using Equation 8)

$$D_i^* = M_i \partial + \bigcup_i \text{ with } \begin{cases} 1 \ if \ D_{i,DSR}^* > D_{i,PTR}^* \\ \\ 0 \ if \ D_{i,DSR}^* < D_{i,PTR}^* \end{cases} \tag{8}$$

The vectors $A_i$ include farm and farmer's socioeconomic and demographic attributes, $\partial$ is the vector of parameters to be measured, $\bigcup_i$ is a random error. In the following step, using the findings of the criterion function, we specify two regime equations that explain the outcome variable of interest. The link between a vector of explanatory variables $W$ and the outcome $P$ may be expressed as $P = f(W)$. More precisely, there are two distinct regimes (Equation 9):

$$Regime \ 1: \ P_{i, DSR} = W_i \gamma + \rho_{i, DSR}, \quad if \ D_i = 1$$

$$Regime \ 1: \ P_{i, PTR} = W_i \delta + \rho_{i, PTR}, \quad if \ D_i = 0 \tag{9}$$

Where $\gamma$ and $\delta$ are parameters to be measured. Moreover, variables under $M_i$ and $W_i$ are allowed to overlap, and error terms $\bigcup_i, \rho_{i, DSR}, \ and \ \rho_{i, PTR}$ have tri-variate normal distribution with zero mean and non-singular covariate matrix [35]. The absence of simultaneous observation of Regime 1 and Regime 2 results in an undefined covariance between $\rho_{i, DSR}$, and $\rho_{i, PTR}$. Furthermore, the expected values of $\rho_{i, DSR}$, and $\rho_{i, PTR}$ are not zero due to the correlation between the error term of the selection Equation 8. We assume that $\Delta_{\bigcup}^2 = 1$ ($\partial$ is estimable only up to scalar). $\Delta_{DSR}^2$ and $\Delta_{PTR}^2$ are variances of the disturbance term used in Equation 9. Moreover, $\Delta_{\bigcup \rho DSR}$ and $\Delta_{\bigcup \rho PTR}$ are covariance of $\bigcup_i$ and $\rho_{i, DSR}$ and covariance of $\bigcup_i$ and $\rho_{i, PTR}$, respectively. The correlation between the error terms in the selection Equation (8) and the regime Equation (9), which are assessed as truncated error terms, causes the expected values of the error terms in Equation (9) to be non-zero [35]. The expected value is calculated as the product of the variance and Inverse Mills Ratios (IMRs) assessed at $M_i^\partial$, denoted as $\aleph_{DSR}$ and $\aleph_{PTR}$, respectively, in Equations 10 and 11.

$$A = \left(p_{i,DSR} D = 1\right) = \Delta_{\bigcup \rho \ DSR} \frac{\phi\left(\frac{M_\partial^i}{\Delta_\bigcup^2}\right)}{\vartheta\left(\frac{M_\partial^i}{\Delta_\bigcup^2}\right)} = \Delta_{\bigcup \rho DSR} \ \aleph_{DSR} \tag{10}$$

$$A = \left(p_{i,PTR} D = 0\right) = \Delta_{\bigcup \rho \ PTR} \frac{\phi\left(\frac{M_\partial^i}{\Delta_\bigcup^2}\right)}{\vartheta\left(\frac{M_\partial^i}{\Delta_\bigcup^2}\right)} = \Delta_{\bigcup \rho PTR} \ \aleph_{\rho PTR} \tag{11}$$

The ESR model can be implemented using a two-stage approach, where regime equations incorporate IMRs. In our investigation, we employ the full information maximum likelihood technique described by Lokshin & Sajaia [36]. For the ESR model to be accurately stated, the factors influencing the selection equation (Equation 8) must include at least one instrument. Additionally, the factors influencing the outcome variables in equation (Equation 9) should be correlated with the adoption of DSR but not directly correlated with the outcome variables. The ESR model's conditional expectations compare the expected outcome variables of DSR adopters and non-adopters in four hypothetical counterfactual circumstances. These cases involve scenarios where adopter smallholders did not adopt, and non-adopter smallholders

did adopt DSR. According to Noltze et al. [37], the following four scenarios can be summarized as:

DSR households with adoption (observed):

$$A\left(P_{DSR}D=1\right)=W'\gamma+\Delta_{\cup\rho DSR}\,\aleph_{\rho DSR} \tag{12a}$$

DSR households without DSR adoption (counterfactual):

$$A\left(P_{PTR}D=1\right)=W'\delta+\Delta_{\cup\rho PTR}\,\aleph_{\rho DSR} \tag{12b}$$

PTR households without adoption (observed):

$$A\left(P_{PTR}D=0\right)=W'\delta+\Delta_{\cup\rho PTR}\,\aleph_{\rho PTR} \tag{12c}$$

PTR households with adoption (counterfactual):

$$A\left(P_{DSR}D=0\right)=W'\gamma+\Delta_{\cup\rho DSR}\,\aleph_{\rho PTR} \tag{12d}$$

In addition to the marginal impacts of $W$ on crop yield and cost of cultivation, we are aiming to determine the treatment impact of DSR adoption. According to Greene [30], Fuglie and Bosch [35], and Alene and Manyong [38] equations 12(a)–12(b) can be utilized to calculate the overall impact of adopting DSR (average treatment effect on the treated, or ATT) and the average treatment effects on those who did not adopt DSR, i.e., the average treatment effect on untreated (ATU). Moreover, we can obtain ATT and ATU from Equation 13 and Equation 14.

$$ATT=A\left(P_{DSR}D=1\right)-A\left(P_{PTR}D=1\right) \tag{13}$$

$$ATU=A\left(P_{DSR}D=0\right)-AA\left(P_{PTR}D=0\right) \tag{14}$$

## Ethics statement

Face-to-face questionnaire surveys were implemented at the participants' residences or fields to collect the data. Verbal consent was obtained from all respondents before the interview. Verbal consent was obtained for two reasons: First, it was anticipated that a substantial number of the respondents would be illiterate and apprehensive about signing consent documents. Secondly, the socioeconomic survey did not collect biological data or tissues or human samples for clinical trials. The interviewers clarified the procedure and emphasized that the subjects' participation was voluntary and that the data collected would be used for research purposes.

## Data

The data used in this study is derived from the households of farmers surveyed in Uttar Pradesh (UP), Madhya Pradesh (MP), Andhra Pradesh (AP), and Telangana State (TS), the four major rice-producing states in India. In these states, labor availability is among the lowest compared to other states, so the DSR method may prove viable and practical. A multi-stage sampling procedure (from state to district to village to farmers) was implemented for the face-to-face questionnaire survey. A survey was conducted on 537 farmers from four different states. Certain surveyed farmers implemented DSR and PTR on distinct plots, whereas

others exclusively utilized DSR or PTR on their respective plots. We take each farm or plot as a separate establishment for the analysis done in the paper. In UP, there were 155 DSR plots and 197 PTR plots; in MP, there were 162 DSR plots and 141 PTR plots; in AP, there were 133 DSR plots and 217 PTR plots; while in TS, there were 197 DSR plots and 205 PTR plots. Our plot-level analysis for yield and cost estimation has 1407 observations. Fig 1 shows that, across the location, among the surveyed households, 46.5–56% of farmers have adopted the DSR practice for producing paddy in the Kharif season.

## Results

### Descriptive statistics

The results for descriptive statistics for socioeconomic variables show that higher education, access to extension services (like agricultural officer/expert visits, agricultural university and Krishi Vigyan Kendra and Krishi Mela services), and community organization membership promote DSR adoption in all four states, regardless of age (Table 2). Nonetheless, if irrigation is not an issue, farmers choose PTR. The DSR approach (Rs.719.5/acre) is 79% less expensive than the PTR method (Rs.3420.5/acre) for land preparation, irrespective of location, owing to reduced tillage operations. The costs of establishing DSR and PTR crops (sowing and transplanting) differ significantly by region. DSR seeding incurs a cost of Rs.1260 per acre, almost one-third of the Rs.3787 per acre required for PTR transplanting. DSR incurs reduced seed and seed treatment expenses (43.5%) compared to PTR owing to diminished seed rates. DSR incurs higher expenses for weeding and pest management than PTR. A 10% increase in weed control cost in DSR enhances paddy yield by 2.72%, 2.82%, 2.83%, and 3.01% in UP, MP, AP, and TS, respectively. However, a similar increase in weed control costs in PTR results in a 1.2%, 1.0%, 1.01%, and 1.1% increase in paddy yield in UP, MP, AP, and TS, respectively. Nevertheless, DSR weed and pest management expenses exceed PTRs by 6.4% and 19.2%, respectively. DSR plots require less family and hired labor (13.4 and 29.3 days/acre) than PTR plots (15.8 and 35.7 days/acre). DSR is less expensive to cultivate than PTR in UP (23.45%), MP (33.1%), AP (19.4%), and TS (38.7%). The 10% increase in fertilizer cost enhances the paddy yield by 1.44%, 2.12%, 3.51%, and 3.66% in DSR. Cost savings are maximized in land preparation, sowing/transplanting, seed rate, and seed treatment. Furthermore, our results show that farmers prioritize yield when selecting new technologies or varieties in areas of moderate production. DSR plots produce 1777 kg/acre less in MP than TPR plots, which yield 2043 kg/acre. In UP, AP, and TS, paddy yields increased by 3.1%, 5.7%, and 6.4% in DSR compared to TPR.

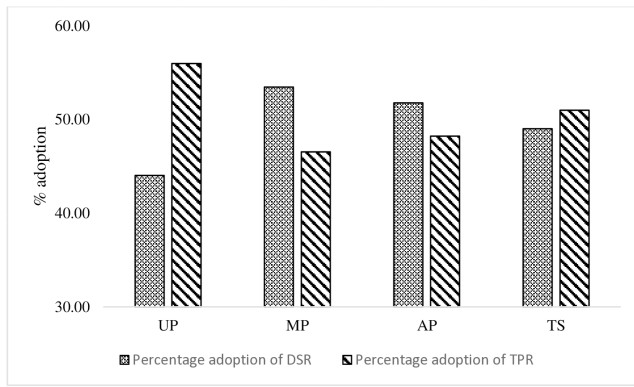

**Fig 1. Percentage of surveyed farmers adopted DSR and PTR in different project locations.**

**Table 2. Descriptive statistics and mean comparison of DSR and PTR in four different states in India.**

| | Uttar Pradesh | | | | | Madhya Pradesh | | | | | Andhra Pradesh | | | | | Telangana | | | | |
|---|---|---|---|---|---|---|---|---|---|---|---|---|---|---|---|---|---|---|---|---|
| | DSR | | PTR | | p-value | DSR | | PTR | | p-value | DSR | | PTR | | p-value | DSR | | PTR | | p-value |
| *Socioeconomic variables* | Mean | Std. Dev | Mean | Std. Dev | | Mean | Std. Dev | Mean | Std. Dev | | Mean | Std. Dev | Mean | Std. Dev | | Mean | Std. Dev | Mean | Std. Dev | |
| Age of house-hold head | 42.61 | 14.26 | 44.29 | 14.11 | 0.191 | 39.71 | 12.54 | 41.13 | 14.31 | 0.142 | 45.33 | 15.82 | 46.64 | 15.41 | 0.116 | 44.31 | 11.16 | 43.12 | 12.36 | 0.162 |
| Education level (Years) | 8.43 | 2.27 | 6.12 | 2.21 | 0.131 | 9.11 | 317 | 7.89 | 4.19 | 0.243 | 10.53 | 2.23 | 8.17 | 1.92 | 0.088 | 11.42 | 2.83 | 9.76 | 3.23 | 0.082 |
| Household size | 5.17 | 2.04 | 5.32 | 2.11 | 0.224 | 4.89 | 1.87 | 5.33 | 2.19 | 0.173 | 4.47 | 1.73 | 4.75 | 1.89 | 0.153 | 4.33 | 1.58 | 5.73 | 1.55 | 0.123 |
| Community organiza-tion(s) mem-ber (%) | 27.13 | 4.18 | 22.56 | 3.28 | 0.162 | 33.17 | 3.43 | 26.19 | 3.11 | 0.093 | 44.72 | 4.19 | 33.17 | 3.77 | 0.083 | 48.13 | 5.54 | 39.37 | 5.18 | 0.092 |
| Farm experience | 21.24 | 6.13 | 23.19 | 5.11 | 0.256 | 18.39 | 6.87 | 20.76 | 5.76 | 0.202 | 21.85 | 6.11 | 22.06 | 7.23 | 0.311 | 22.33 | 5.56 | 23.01 | 6.34 | 0.256 |
| Assured irrigation | 28.37 | 5.17 | 42.17 | 6.92 | 0.042 | 22.39 | 5.89 | 36.17 | 6.15 | 0.073 | 44.18 | 7.13 | 57.19 | 8.71 | 0.074 | 66.15 | 8.17 | 83.12 | 8.33 | 9.46 |
| Livestock ownership (%) | 0.64 | 0.18 | 0.68 | 0.16 | 0.222 | 0.58 | 0.17 | 0.6 | 0.14 | 0.188 | 0.61 | 0.22 | 0.67 | 0.19 | 0.223 | 0.48 | 0.17 | 0.5 | 0.21 | 0.187 |
| Smartphone ownership | 0.66 | 0.18 | 0.57 | 0.11 | 0.112 | 0.63 | 0.23 | 0.58 | 0.17 | 0.222 | 0.75 | 0.11 | 0.72 | 0.13 | 0.232 | 0.78 | 0.15 | 0.72 | 0.13 | 0.183 |
| Extension service | 18.42 | 3.67 | 14.12 | 3.51 | 0.184 | 21.73 | 4.66 | 17.82 | 3.47 | 0.215 | 36.58 | 5.22 | 31.82 | 4.51 | 0.193 | 39.16 | 5.72 | 32.36 | 3.95 | 0.129 |
| Crop insurance | 12.79 | 2.46 | 11.87 | 2.65 | 0.246 | 15.16 | 2.37 | 14.63 | 2.56 | 0.261 | 17.97 | 3.27 | 15.83 | 3.27 | 0.213 | 19.77 | 3.45 | 17.46 | 2.89 | 0.221 |
| Total land-holding (acre) | 1.63 | 0.42 | 1.69 | 0.39 | 0.154 | 2.43 | 0.54 | 2.51 | 0.51 | 0.243 | 1.73 | 0.83 | 1.68 | 0.73 | 0.202 | 2.46 | 1.02 | 2.54 | 0.93 | 0.197 |
| *Outcome variables* | | | | | | | | | | | | | | | | | | | | |
| Land prepara-tion cost (Rs/acre) | 556 | 83.23 | 3606 | 346.17 | 0.014 | 876 | 103.72 | 2634 | 264.83 | 0.012 | 777 | 123.92 | 2402 | 279.35 | 0.008 | 669 | 92.37 | 5040 | 456.81 | 0.004 |
| Seed and seed treatment cost | 959 | 58.42 | 1248 | 76.39 | 0.043 | 1017 | 114.57 | 2510 | 254.39 | 0.032 | 866 | 115.23 | 1397 | 165.76 | 0.036 | 827 | 89.17 | 1329 | 145.74 | 0.027 |
| Sowing/Trans-planting cost | 1151 | 231.54 | 2773 | 335.83 | 0.004 | 1498 | 245.91 | 5198 | 554.76 | 0.001 | 1445 | 234.17 | 3615 | 337.89 | 0.003 | 945 | 465.46 | 3563 | 654.17 | 0.002 |
| Total fertilizer cost (Rs/acre) | 2286 | 321.16 | 2607 | 455.83 | 0.137 | 2776 | 355.18 | 3678 | 448.27 | 0.082 | 3155 | 376.19 | 3113 | 418.62 | 0.167 | 3854 | 444.17 | 4261 | 537.87 | 0.143 |
| Irrigation cost (Rs/acre) | 1754 | 273.67 | 2348 | 389.45 | 0.112 | 816 | 289.14 | 1549 | 333.54 | 0.054 | 1768 | 334.28 | 2715 | 476.17 | 0.065 | 1715 | 287.34 | 2874 | 322.19 | 0.043 |
| Weed control cost (Rs/acre) | 1646 | 209.17 | 1478 | 188.76 | 0.167 | 1432 | 333.18 | 1235 | 342.18 | 0.152 | 1599 | 245.19 | 1781 | 311.54 | 0.132 | 2054 | 198.62 | 1834 | 202.41 | 0.156 |
| Pest control cost | 1680 | 265.16 | 1297 | 176.54 | 0.108 | 1216 | 265.17 | 1052 | 211.45 | 0.167 | 2245 | 372.74 | 1879 | 187.54 | 0.034 | 2000 | 244.87 | 1763 | 221.34 | 0.109 |

*(Continued)*

**Table 2.** (Continued)

| Socioeconomic variables | Uttar Pradesh | | | | | Madhya Pradesh | | | | | Andhra Pradesh | | | | | Telangana | | | | |
|---|---|---|---|---|---|---|---|---|---|---|---|---|---|---|---|---|---|---|---|---|
| | DSR | | PTR | | p-value | DSR | | PTR | | p-value | DSR | | PTR | | p-value | DSR | | PTR | | p-value |
| | Mean | Std. Dev | Mean | Std. Dev | | Mean | Std. Dev | Mean | Std. Dev | | Mean | Std. Dev | Mean | Std. Dev | | Mean | Std. Dev | Mean | Std. Dev | |
| Harvesting cost | 2490 | 333.87 | 2496 | 287.51 | 0.243 | 1763 | 436.73 | 1800 | 387.19 | 0.237 | 2774 | 224.45 | 2759 | 276.38 | 0.221 | 1542 | 278.87 | 2346 | 299.16 | 0.067 |
| Post-harvest cost | 6426 | 1176 | 6872 | 1380 | 0.187 | 5992 | 1083 | 6332 | 1265.34 | 0.143 | 7592 | 1065 | 7894 | 1285 | 0.254 | 3037 | 556.87 | 4144 | 455.9 | 0.081 |
| Total labor (days/acre) | 27.16 | 3.67 | 35.15 | 4.76 | 0.052 | 28.31 | 4.87 | 35.18 | 5.87 | 0.243 | 28.55 | 3.45 | 36.16 | 4.27 | 0.057 | 33.18 | 2.34 | 36.16 | 2.76 | 0.116 |
| Family labor (days/acre) | 14.13 | 2.64 | 16.67 | 3.08 | 0.176 | 13.28 | 3.17 | 16.69 | 3.18 | 0.183 | 12.77 | 2.18 | 14.21 | 1.98 | 0.132 | 13.33 | 1.83 | 15.75 | 1.65 | 0.211 |
| Total cost (Rs/acre) | 18948 | 2189 | 24725 | 3546 | 0.043 | 17386 | 4567 | 25988 | 5432 | 0.035 | 22221 | 3786 | 27555 | 4111 | 0.069 | 16643 | 3546 | 27154 | 4467 | 0.005 |
| Rice yield (kg/acre) | 1793 | 246.87 | 1739 | 254.19 | 0.219 | 1777 | 143 | 2043 | 154 | 0.046 | 2666 | 146 | 2523 | 231 | 0.443 | 2523 | 118 | 2371 | 107 | 0.172 |
| Income from rice (Rs/acre) | 20193 | 3467 | 13237 | 3142 | 0.062 | 22241 | 4653 | 19162 | 4576 | 0.082 | 40217 | 4776 | 34883 | 3987 | 0.072 | 48955 | 5671 | 33306 | 4782 | 0.001 |
| Rice selling price (Rs/Quintal) | 2183 | 45.64 | 2183 | 38.65 | 0.165 | 2230 | 35.76 | 2210 | 45.82 | 0.215 | 2342 | 28.97 | 2342 | 32.45 | 0.177 | 2600 | 26.8 | 2550 | 33.18 | 0.119 |

## Impact of DSR-practice adoption

**Quality of matching using a covariate balancing test.** Prior to evaluating the effects of DSR adoption, it is necessary to ensure that the distribution of relevant variables is balanced between adopters of the DSR establishment technique and non-adopters. The comprehensive outcomes of the covariate balancing test for each state, both pre- and post- matching are presented in Table 3. In PSM, the standardized mean difference for all covariates is decreased by 32% to 46%, regardless of the matching algorithm employed. It signifies a significant decrease in overall bias achieved via matching. Furthermore, upon matching, the ρ-values of the likelihood ratio tests indicate that the joint significance of covariates is rejected, whereas it was not rejected before matching. Following matching, the pseudo-$R^2$ values for all three matching algorithms decrease substantially, irrespective of the states. The low pseudo-$R^2$, insignificant ρ-values of the likelihood ratio test, and low mean standardized bias suggest that the propensity score effectively achieves a state of equilibrium in the distribution of covariates between the two groups (DSR adopters and non-adopters).

**Impact assessment.** The average impact of DSR adoption in different geographies (UP, MP, AP, and TS) is estimated using the PSM method. The study uses three matching algorithms, i.e., nearest neighbor matching (NNM), kernel-based matching (KBM), and radius matching (RM), to measure the average treatment effect on treated (ATT). ATT measures the difference in outcomes between DSR adopters and non-adopters. S1-S4 Tables represents the impact of DSR adoption on the cost of cultivation, yield, and income for the UP, MP, AP, and TS, respectively. The results support our findings and inferences in descriptive analysis. Table 4 shows the comparative impact of DSR adoption in 4 different geographies. The results indicate that land preparation savings are highest in TS (Rs3480/acre) and lowest in AP (Rs1586/acre). The seed and seed treatment cost is also saved (Rs279/acre to Rs545/acre based on the location) in DSR. This trend is found in all four geographies. In PTR, the seed rate (25–30 kg/acre) is 2.5 to 3 times higher than DSR (8–10 kg/acre). Hence, farmers invest more in PTR seed costs than DSR.

Farmers save the crop establishment cost of Rs1580/acre to Rs3172/acre based on the location by adopting DSR. We observed that the saving is maximum in MP and lowest in UP. Nevertheless, the cultivation cost in PTR exceeds that of DSR irrespective of geographies. The additional savings in crop establishment are higher in TS and AP than in UP and MP. In all the

Table 3. Covariate balancing test before and after matching.

| | | Matching algorithm | Pseudo-R2 before matching | Pseudo-R2 after matching | ρ> χ2 before matching | ρ> χ2 after matching | Mean standardized bias before matching | Mean standardized bias after matching | (Total)% \|bias\| reduction |
|---|---|---|---|---|---|---|---|---|---|
| Adoption to DSR | Uttar Pradesh | NNM | 0.177 | 0.048 | 0.017 | 0.267 | 33.19 | 20.52 | 38.16 |
| | | RM | 0.165 | 0.044 | 0.024 | 0.283 | 32.56 | 22.09 | 32.15 |
| | | KBM | 0.183 | 0.039 | 0.032 | 0.304 | 35.65 | 20.98 | 41.16 |
| | Madhya Pradesh | NNM | 0.154 | 0.052 | 0.025 | 0.299 | 28.89 | 18.17 | 37.1 |
| | | RM | 0.167 | 0.046 | 0.033 | 0.276 | 29.92 | 18.99 | 36.54 |
| | | KBM | 0.172 | 0.056 | 0.028 | 0.256 | 27.17 | 17.50 | 35.59 |
| | Andhra Pradesh | NNM | 0.155 | 0.036 | 0.022 | 0.299 | 35.26 | 21.71 | 38.44 |
| | | RM | 0.149 | 0.032 | 0.037 | 0.354 | 37.11 | 21.72 | 41.46 |
| | | KBM | 0.143 | 0.033 | 0.031 | 0.324 | 35.71 | 20.04 | 43.89 |
| | Telan-gana | NNM | 0.168 | 0.045 | 0.029 | 0.273 | 33.66 | 18.27 | 45.72 |
| | | RM | 0.162 | 0.041 | 0.03 | 0.318 | 30.45 | 17.16 | 43.65 |
| | | KBM | 0.154 | 0.036 | 0.034 | 0.307 | 29.18 | 17.27 | 40.83 |

locations except MP, fertilizer costs play an insignificant role in the PTR and DSR methods. In MP, adopters of DSR experienced less fertilizer cost of Rs772/acre due to improved nutrient efficiency with split application than DSR non-adopters.

Irrigation costs across all locations are significantly lower in DSR than in PTR. The overall water requirement in DSR is 30–50% less than in PTR, resulting in less irrigation cost in DSR than PTR. Unlike irrigation, the plant protection costs (weed and pest control costs) are higher in DSR. Results also indicate that farmers with DSR adoption can save 9.33 to 11.20 man-days (Hired and Family labor) compared to DSR farmers. Our findings show that the total cost of cultivation is significantly less (Rs5043/acre to Rs7718/acre) in DSR than in conventional PTR farming. Except for MP, an increased yield of 58 kg/acre, 92 kg/acre, and 126 kg/acre under DSR is reported in UP, AP, and TS, respectively. In MP, the yield under DSR is significantly lower (179 kg/acre) than PTR. Nevertheless, results indicate that the net

**Table 4. Comparative analysis of PSM outcomes.**

| Variables | UP | MP | AP | TS |
|---|---|---|---|---|
| | ATT | | | |
| Land preparation cost (Rs/acre) | −2908*** | −1769*** | −1586*** | −3480*** |
| | (47.84) | (49.12) | (60.56) | (128.08) |
| Seed and seed treatment cost (Rs/acre) | −279** | −545** | −533** | −516** |
| | (71.01) | (75.89) | (60.60) | (56.49) |
| Crop establishment cost (Rs/acre) | −1580*** | −1705*** | −2120*** | −1951*** |
| | (222.34) | (244.83) | (266.24) | (140.18) |
| Total fertilizer cost (Rs/acre) | −466 | −772** | −335 | −529 |
| | (83.49) | (110.79) | (51.20) | (67.28) |
| Irrigation cost (Rs/acre) | −629** | −728** | −1092*** | −1203*** |
| | (74.22) | (141.88) | (209.49) | (125.81) |
| Weed control cost (Rs/acre) | 161** | 307** | 241** | 322** |
| | (36.92) | (66.45) | (44.34) | (60.89) |
| Pest control cost | 398** | 37.33* | 294** | 207** |
| | (61.27) | (17.47) | (69.39) | (67.96) |
| Harvesting cost | −58.67 | −37.67 | −115.00 | −317.67 |
| | (56.09) | (23.51) | (78.90) | (42.84) |
| Post-harvest cost | −104.00 | −345.00 | −308.33 | −722.00 |
| | (78.80) | (166.04) | (87.87) | (156.87) |
| Total labor (days/acre) | −8.02*** | −6.92*** | −7.64*** | −7.70*** |
| | (1.04) | (1.10) | (1.19) | (0.40) |
| Family labor (days/acre) | −3.18** | −3.15** | −1.69** | −2.69** |
| | (0.34) | (0.48) | (0.39) | (0.28) |
| Total cost (Rs/acre) | −6867*** | −5552*** | −5043*** | −7717*** |
| | (679.67) | (447.44) | (500.08) | (436.37) |
| Rice yield (kg/acre) | 58 | −178** | 92.33** | 125.67** |
| | (48.28) | (78.06) | (30.59) | (36.35) |
| Income from rice (Rs/acre) | 6238*** | 5693*** | 5009*** | 8134*** |
| | (578.33) | (388.71) | (424.23) | (1293.67) |

*= significant at 10%;

**= significant at 5%;

***= significant at 1%.

income from paddy cultivation using the DSR method is significantly higher, irrespective of different geographies. For states like UP, MP, and AP, the net income from DSR adoption varies between Rs5009/acre to Rs6237/acre, while in TS, the net income from DSR adoption is Rs 8133/acre higher than PTR.

**ESR-based paddy yield impact.** The findings identified that the PSM technique may be biased due to unobservable variables. Consequently, this investigation has implemented the FILM-based ESR model to mitigate the inherent biases and reliability issues linked to the PSM model. Table 5 illustrates the average treatment effect of DSR adoption on paddy yield in both actual and counterfactual scenarios. The data is derived from the ESR method. Compared to non-DSR adopters, DSR adopters attain increased yields in UP, AP, and TS. However, conventional PTR farmers of MP achieve better yields (10.69%) than DSR farmers. In UP, AP, and TS, if DSR non-adopters decide to adopt DSR, their yield can improve by 3.58%, 3.15%, and 4.49%, respectively.

Table 6 provides parameter estimates for the yield effects of the DSR adoption on a double-log specification of the Production Function. The Cobb-Douglas specification could not be rejected, as indicated by the Wald test in the lower panel of Table 6. The estimated covariance terms are also displayed in the bottom portion of Table 6. Statistical analysis indicates heterogeneity, which could result in biased estimates if not corrected. The IMR coefficient is negative, indicating that the estimates would have been downward biased if the

**Table 5. Average treatment effects of DSR on rice yield, cost of cultivation, and net income in four different states in India.**

| Variables | Type of farmers and treatment effect | Uttar Pradesh | | | Madhya Pradesh | | | Andhra Pradesh | | | Telangana | | |
|---|---|---|---|---|---|---|---|---|---|---|---|---|---|
| | | Decision stage | | ATEs | Decision stage | | ATEs | Decision stage | | ATEs | Decision stage | | ATEs |
| | | To adopt DSR farming | Not to adopt DSR farming | | To adopt DSR farming | Not to adopt DSR farming | | To adopt DSR farming | Not to adopt DSR farming | | To adopt DSR farming | Not to adopt DSR farming | |
| Yield (kg/acre) | DSR adopting farmers | 1775 | 1718 | 57 | 1806 | 1986 | −180** | 2671 | 2583 | 88* | 2563 | 2457 | 106* |
| | DSR non-adopting farmers | 1764 | 1703 | 61 | 1823 | 1999 | −176** | 2654 | 2573 | 81** | 2555 | 2445 | 110* |
| | Heterogeneous effect | 11 | 15 | −4 | −17 | −13 | −4 | 17 | 10 | 7 | 8 | 12 | −4 |
| Cost of cultivation | DSR adopting farmers | 18251 | 25339 | −7088*** | 17513 | 23105 | −5592*** | 22347 | 27104 | −4757*** | 17422 | 24932 | −7510*** |
| | DSR non-adopting farmers | 18443 | 25603 | −7160*** | 17654 | 23448 | −5794*** | 22587 | 27325 | −4738*** | 17804 | 25265 | −7461*** |
| | Heterogeneous effect | −192 | −264 | 72 | −141 | −343 | 202 | −240 | −221 | −19 | −382 | −333 | −49 |
| Net income | DSR adopting farmers | 19706 | 14036 | 5670*** | 22521 | 17103 | 5418*** | 39115 | 34753 | 4362*** | 42634 | 34659 | 7975*** |
| | DSR non-adopting farmers | 19432 | 13803 | 5629*** | 21987 | 16653 | 5334*** | 38606 | 34345 | 4261*** | 42217 | 34387 | 7830*** |
| | Heterogeneous effect | 274 | 233 | 41 | 534 | 450 | 84 | 509 | 408 | 101 | 417 | 272 | 145 |

*=significant at 10%;

**=significant at 5%;

***=significant at 1.

**Table 6. Endogenous switching regression outcomes for paddy yield across four paddy growing states in India.**

| Variables | Uttar Pradesh | | | Madhya Pradesh | | | Andhra Pradesh | | | Telangana | | |
|---|---|---|---|---|---|---|---|---|---|---|---|---|
| | Criterion function | Regime equation | | Criterion function | Regime equation | | Criterion function | Regime equation | | Criterion function | Regime equation | |
| | | DSR | Conventional PTR | DSR | DSR | Conventional PTR | DSR | DSR | Conventional PTR | Conventional PTR | DSR | Conventional PTR |
| Land preparation cost (Rs/acre), log | −0.662** (0.218) | 0.068 (0.052) | 0.127** (0.043) | −0.615** (0.226) | 0.143** (0.041) | 0.073 (0.056) | −0.585** (0.342) | 0.167** (0.055) | 0.071* (0.021) | −0.666 (0.378) | 0.176* (0.066) | 0.112** (0.054) |
| Seed and seed treatment cost (Rs/acre), log | −0.411** (0.156) | −0.072** (0.019) | 0.056 (0.018) | −0.781 (0.245) | 0.078 (0.033) | −0.092** (0.056) | −0.693 (0.331) | 0.061 (0.031) | −0.112** (0.039) | −0.558 (0.121) | −0.121** (0.049) | 0.045** (0.021) |
| Crop establishment cost (Rs/acre), log | −0.217 (0.167) | 0.031 (0.023) | 0.043 (0.032) | −0.229 (0.154) | 0.042 (0.038) | 0.027 (0.022) | −0.231 (0.176) | 0.031 (0.026) | 0.022 (0.019) | −0.245 (0.089) | 0.021 (0.016) | 0.033 (0.024) |
| Total fertilizer cost (Rs/acre), log | −0.539** (0.222) | 0.144** (0.041) | 0.181** (0.054) | −0.482 (0.255) | 0.241** (0.079) | 0.212** (0.063) | −0.462 (0.243) | 0.391** (0.107) | 0.351** (0.101) | −0.473 (0.253) | 0.366** (0.118) | 0.411** (0.128) |
| Irrigation cost (Rs/acre), log | −0.714** (0.298) | 0.187*** (0.064) | 0.265*** (0.089) | −0.726 (0.262) | 0.247*** (0.088) | 0.172*** (0.044) | −0.679 (0.233) | 0.337*** (0.099) | 0.234*** (0.073) | −0.688 (0.241) | 0.229*** (0.097) | 0.328*** (0.134) |
| Weed control cost (Rs/acre), log | −0.629*** (0.331) | 0.272*** (0.079) | 0.116** (0.043) | −0.635 (0.335) | 0.104** (0.086) | 0.287*** (0.078) | −0.661 (0.399) | 0.114*** (0.037) | 0.283*** (0.087) | −0.614 (0.415) | 0.301*** (0.124) | 0.109** (0.054) |
| Pest control cost (Rs/acre), log | −0.547* (0.246) | 0.087 (0.058) | 0.093* (0.039) | −0.543 (0.191) | 0.114* (0.052) | 0.107 (0.071) | −0.528 (0.272) | 0.084* (0.044) | 0.076 (0.069) | −0.544 (0.321) | 0.118 (0.069) | 0.126* (0.067) |
| Total labor cost (Rs/acre), log | −0.436** (0.183) | 0.013 (0.011) | 0.227** (0.079) | −0.449 (0.262) | 0.224** (0.086) | 0.011 (0.008) | −0.417 (0.233) | 0.218** (0.089) | 0.012 (0.007) | −0.439 (0.241) | 0.014 (0.011) | 0.222** (0.032) |
| Age of household head | 0.028 (0.024) | | | 0.032 (0.026) | | | 0.022 (0.021) | | | 0.025 (0.021) | | |
| Education level (Years) | 0.016** (0.004) | | | 0.018** (0.007) | | | 0.025** (0.008) | | | 0.029** (0.011) | | |
| Household size | 0.015 (0.011) | | | 0.009 (0.006) | | | −0.014** (0.006) | | | 0.007 (0.006) | | |
| Community organization(s) member (%) | 0.033** (0.012) | | | 0.027** (0.013) | | | 0.031** (0.011) | | | 0.033** (0.011) | | |
| Farm experience | 0.013 (0.008) | | | 0.015 (0.011) | | | 0.015 (0.009) | | | 0.015* (0.007) | | |
| Assured irrigation | −0.026** (0.011) | | | −0.022** (0.009) | | | −0.018** (0.006) | | | −0.037** (0.009) | | |

*(Continued)*

**Table 6.** (Continued)

| Variables | Uttar Pradesh | | | Madhya Pradesh | | | Andhra Pradesh | | | Telangana | | |
|---|---|---|---|---|---|---|---|---|---|---|---|---|
| | Criterion function | Regime equation | | Criterion function | Regime equation | | Criterion function | Regime equation | | Criterion function | Regime equation | |
| | | DSR | Conventional PTR | DSR | DSR | Conventional PTR | DSR | DSR | Conventional PTR | Conventional PTR | DSR | Conventional PTR |
| Livestock ownership (%) | 0.018 | | | 0.015 | | | 0.013 | | | 0.015 | | |
| | (0.013) | | | (0.011) | | | (0.011) | | | (0.011) | | |
| Smartphone ownership | 0.024 | | | 0.027 | | | 0.024 | | | 0.027 | | |
| | (0.02) | | | (0.022) | | | (0.020) | | | (0.022) | | |
| Extension service | 0.037** | | | 0.017* | | | 0.025* | | | 0.028* | | |
| | (0.012) | | | (0.006) | | | (0.008) | | | (0.007) | | |
| Crop insurance | 0.019** | | | 0.008 | | | 0.014** | | | 0.018 | | |
| | (0.004) | | | (0.004) | | | (0.004) | | | (0.014) | | |
| Total landholding (acre) | 0.017 | | | 0.028** | | | 0.016 | | | 0.016** | | |
| | (0.013) | | | (0.007) | | | (0.014) | | | (0.007) | | |
| Constant | −6.116 | 4.165 | 5.769 | −7.265 | 5.934 | 4.339 | −8.413 | 6.004 | 5.091 | −7.534 | 5.567 | 4.659 |
| LR (Wald test) for independent equations | 1.634 187.18 (p-value less than 0.001) | | | | | | | | | | | |

*=significant at 10%;

**=significant at 5%;

***=significant at 1.

correction had not been made. In addition to the inputs used in rice cultivation, the analysis considers variations in several socio-economic and demographic explanatory variables (e.g., age, farm experience, assured irrigation facility, education of the decision maker, etc.) in our criterion (selection) model, to minimize the biases in outcomes. The criterion function (selection equation in the model) and the regime equations are concurrently estimated in the ESR method to determine whether an observation belongs to a particular regime. The criterion equation includes all explanatory and instrumental variables [36]. In this study, we employed 11 socio-economic and demographic variables (Table 6) as instruments in the criterion function.

The parameter estimates of the criterion function are presented in Table 6. Costs associated with land preparation, seed and seed treatment, crop establishment, irrigation, weed control, fertilizer, and labor costs are the significant factors influencing the adoption of the DSR method across different geographies (UP, MP, AP, and TS). Furthermore, the adoption of DSR is significantly influenced by factors such as the farmer's education, community organization membership, assured irrigation, and extension service.

The outcomes indicate that in the DSR method, the weed control cost has the highest positive elasticity. A 10% increase in weed control cost enhances paddy yield by 2.72%, 2.82%, 2.83%, and 3.01% in UP, MP, AP, and TS, respectively. The cost of weed control also has a positive, albeit minor, impact on rice production under PTR. A 1% increase in weed control costs results in a 0.12%, 0.10%, 0.11%, and 0.11% increase in paddy yield in UP, MP, AP, and TS. Like weed, irrigation cost also shows a positive elasticity with paddy productivity across all states. Similarly, the fertilizer cost shows positive elasticity with yield across all the locations. A 10% increase in fertilizer cost enhances the paddy yield by 1.44%, 2.12%, 3.51%, and 3.66%. In AP and TS, the land preparation costs in DSR also show positive elasticity with paddy yield: a 1% increase in land preparation cost augments the productivity of paddy by 0.07% and 0.18% in AP and TS. The DSR method exhibits positive elasticity for total labor: a 1% increase in total labor cost results in a rise in paddy yield of 0.22%–0.23% across all geographies.

## Discussion

Farmers prefer DSR over PTR because of reduced cultivation expenses. However, increased weed pressure and the necessity for more frequent irrigation sometimes diminish its attractiveness. Lack of capacity building among farmers, including low educational attainment and limited access to extension services, may explain MP's lower yield under DSR adoption [39,40]. However, the paddy yield increases with DSR adoption in UP, AP, and TS. Several South-Asian studies support this finding [30,39,41–43]. According to these studies, puddling in PTR can lead to loss of soil structure and make the soil compact, negatively affecting root growth and water infiltration. Conversely, in DSR, the plant's root growth is more profound and stronger as they are not confined to the hardpan. This can lead to better nutrient uptake, overall plant health, and increased yield. Additionally, the practice of alternate wetting and drying irrigation methods in DSR enhances microbial growth within the plant root environment due to the oxygenated rhizosphere. The interaction between microbes and plant roots further boosts the plant's access to nutrients, resulting in improved biological yield compared to PTR. Conversely, maintaining flooded conditions in fields restricts the growth of aerobic microorganisms in the rhizosphere.

Besides crop physiology and nutrient dynamics, other agronomic practices influence farmers' decision to adopt DSR. The parameter estimates of the criterion function indicate that costs related to land preparation, seed and seed treatment, crop establishment, irrigation, weed control, fertilizer, and labor are among the significant factors influencing the adoption

of the DSR across the geographies. Irrigation cost exhibits a positive elasticity with paddy productivity across all states. Optimum irrigation in DSR plots based on crop requirements and water availability in soil contributes to increased paddy yield. Frequent irrigation mitigates crop water stress and reduces weed growth effectively. Our observations in this study comparing different geographies indicate that the capacity of soil to retain water strongly influences the decisions for determining the quantity and timing of irrigation. UP and MP possess sandy loam and black soil, which exhibit enhanced water-holding capacity. Consequently, these regions have reduced irrigation culture compared to AP and TS, which feature red lateritic soil with lower water-holding capacity. Therefore, compared to UP and MP, an increase in irrigation adds to input costs but gets compensated by the likelihood of paddy yield in AP and TS. However, if we see the conventional PTR, intensive land preparation activities, including primary and secondary tillage, puddling, harrowing, and leveling, in order to maintain a ponded water level of 3–5 cm for a duration of up to 90–100 days (for medium duration paddy variety) enhances the land preparation and irrigation costs. While DSR adopters apply intermittent water through alternate wetting and drying, depending on soil moisture conditions. Thereby, the total water requirement in DSR is 30–50% lower than that of PTR, leading to reduced irrigation costs in DSR compared to PTR.

Overall, the costs of different intercultural operations compared to the improvement in crop yield are the fundamental determinants for adoption. If the addition of costs results in significant improvement in grain yields, farmers may invest upfront. However, while conducting this study, we observed a peculiar case where the crop yield under DSR dropped considerably more than PTR in the state of MP. This can be attributed to compacted heavy soil conditions resulting in complications such as delayed emergence, an increased risk of seedling diseases, and root rot. The drudgery involved growing the nursery on a designated piece of land, transporting the nursery to various plots, and transplanting the seedlings, further enhancing the cost of paddy cultivation PTR. The DSR method does not necessitate extensive land preparation activities, which have both positive and negative aspects. The minimal land preparation and lack of transplanting activities in DSR contribute to significant savings in working hours and costs for hired and family laborers. However, the absence of intensive tillage activities and the implementation of alternate wetting-drying irrigation practices contribute to elevated weed pressure in DSR. Consequently, both pre-emergence and post-emergence weedicides are applied in DSR plots to protect the seedlings from weed competition and nutrient depletion. Pest attacks are also more prevalent in DSR than in PTR. Hence, in DSR, farmers implement measures to safeguard seeds against insect and pest infestations to enhance germination rates and crop yield. Pre-treatment of seeds with insecticides and pesticides increases the overall seed treatment costs associated with DSR. The states in peninsular regions of India can compensate for the costs mentioned above through a markedly higher DSR yield than PTR. Among the four states, DSR is most profitable in TS. The most net return from DSR in TS can be attributed to larger landholdings and enhanced mechanization, effectively lowering labor costs. Additionally, improved market channels and price realization attributed to making DSR more profitable in TS than in other regions.

We also observed that the adoption of DSR is further influenced by socio-economic and demographic variables [36]. Farmers with more formal education are more likely to understand the information provided by agricultural experts and extension agents, enabling them to make well-informed judgments on new technology adoption. In addition, farmers with a formal education are more adept at forecasting the possible consequences of future technologies on agricultural productivity and financial viability. Therefore, there is a positive association between the educational level of the household head and their willingness to adopt innovative and sustainable farming practices, such as DSR. This observation is consistent with

the findings published by Duraisamy [44], Idrisa et al. [45], and Huffman [46]. Adeoti [47] and Nonvide [48] found that farmers with greater levels of education demonstrate improved capacities to adopt and efficiently utilize sustainable technology.

Similarly, membership in a group or organization is a reliable measure of social capital. Social networks facilitate the exchange of knowledge and promote peer-to-peer learning among farmers. Social organizations functioned as an informal means of providing insurance during periods of distress. When group members endorse deploying new technology, it creates a favorable mindset among farmers, facilitating technology acceptance. This could explain why farmers with a group membership are more likely to adopt DSR practices. Farmers also obtain information regarding the most recent technologies through interaction with the extension agent. Farmers can address any uncertainties related to the technology by establishing direct communication with extension personnel. The guidance offered by extension agents on optimizing yields and reducing cultivation costs effectively encourages farmers to adopt DSR in paddy farming.

Assured irrigation negatively affects the adoption of DSR technology. It indicates that farmers with assured access to irrigation choose PTR. The DSR method is primarily designed to conserve water in rice cultivation, minimize methane emissions from paddy fields, and improve soil health. Nevertheless, farmers who have assured irrigation are mostly unaware of the detrimental impacts of flood irrigation on soil and the ecosystem [47]. Furthermore, in some states, the state government offers electricity for irrigation at a reduced cost or for free, which motivates farmers to continue flood irrigation based on the PTR method rather than adopt the water-saving DSR method.

The findings also suggest that an increase in the use of crop insurance has a favorable effect on the adoption of DSR for the farmers in UP and AP. Farmers who adopt DSR instead of conventional PTR also believe that if their crop yield is negatively affected by their lack of proficiency in adopting the new technique, crop insurance will offer an extra layer of protection against the resulting losses. Utilizing a certain risk management instrument encourages the adoption of other risk management solutions.

The results in Table 6 also suggest that landholding size positively impacts the adoption of DSR in MP and TS. This discovery is consistent with the results of prior investigations. The farmers in UP and AP are mostly smallholders. They believe that they cannot attain food security and expected income if they lose their average yield due to adopting new technology. However, in both states, the yield in DSR is higher than that of PTR. Still, the adoption of DSR is low compared to PTR. This is because delay in monsoon causes low seed germination. Sometimes, intense rainfall within 15–20 days after sowing damages both seed and seedlings. In such a scenario, farmers need to invest more in seed procurement and additional labor for gap-filling. Moreover, socio-psychological factors, like the risk of losing yield if they fail to control weeds, risk of social exclusion from the community if they adopt a new practice, and difficulty in machine harvesting as DSR plots mature 10–15 days earlier than PTR, restricts the farmers to adopt DSR. Meanwhile, large landowners in MP and TS allocate their land under both interventions (PTR and DSR). They are confident that they can recoup their income and guarantee food security through PTR despite the potential decrease in yield as a result of the implementation of DSR, given their relatively large farm size.

The intricate linkage of weed control with irrigation scheduling exhibits the highest positive elasticity in DSR adoption. Effective management of weeds, especially during the seedling phase, is done by applying pre and post-emergence herbicides while maintaining the moist conditions for up to 30 days to achieve optimum results with manual weeding whenever required. However, the expense associated with weed control exerts a beneficial, though limited, influence on rice production within the context of PTR. Similarly, the fertilizer cost

exhibits a positive elasticity with yield across the north Indian states, which can be referred to as highly fertile soil compared to AP and TS. Therefore, an increase in the application rate of fertilizers, especially nitrogen-based fertilizers, significantly improves paddy yield in peninsular Indian states.

We also observed that the land preparation costs in DSR exhibit a positive elasticity with crop yield in both AP and TS. Farmers in AP and TS implement permanent bunds, conduct laser land leveling, and occasionally engage in mild puddling to enhance soil water-holding capacity. This is necessitated by the lower water holding capacity and undulating terrain in these regions compared to UP and MP. The DSR method demonstrates a positive elasticity concerning total labor. In DSR, the labor cost exhibits negligible elasticity concerning paddy productivity, indicating that labor productivity rises with the adoption of DSR.

The yield of rice in DSR is further influenced by both the quantity and quality of the seeds chosen by the farmer. Suboptimal seed rate and quality can result in reduced germination rates. Treated seeds, whether through physical, chemical, or biological methods, are preferable for DSR as they protect against avian consumption or damage to the seeds, enhance germination rates, and support early-stage growth. Failure to select quality and suitable seeds under DSR can significantly reduce yields across various geographical regions. Kumar & Ladha [29] and Yadav et al. have determined that an optimal seed rate is essential for achieving optimal paddy production in DSR. Exceeding the optimal seed quantity may result in a decrease in yield. The increased density of seedlings or plants within a given area leads to heightened competition, subsequently reducing overall yield.

## Conclusion

Our study examines the adoption of DSR among farmers in four Indian rice-producing states using the PSM approach. The ESR method measures the effects of selection bias and heterogeneity caused by DSR, allowing for a more comprehensive understanding of its economic suitability in different agro-climatic geographies. We conclude that adopting DSR can lead to variations in paddy yield depending on agro-climatic conditions. However, net income from paddy farming experienced a significant increase regardless of state/agro-climatic conditions. Increased investments in weed management, irrigation, and fertilizer can lead to higher yields in DSR. Socioeconomic variables such as education, community organization membership, extension services, and crop insurance positively impact DSR adoption, while assured irrigation negatively affects it. Our study points out that India has started facing significant challenges in water and manpower availability at critical periods of agricultural operations, which are key to agricultural development, particularly in paddy production. We wish to raise the alarm that the water table is steadily decreasing in many northern and peninsular India regions. Therefore, it is essential to advocate for adopting resource-conserving technologies like DSR to guarantee food security for present and future generations. Policymakers should create provisions for incentives to promote DSR adoption, prioritizing the development of skill sets in weed management through optimal management techniques. Our study picks certain peculiar though important repercussions like the effect of subsidies on power for agricultural uses and the role of the socio-psychological state of mind in the adoption of DSR. For the first time, this study reported both aspects of DSR impact, allowing the audience to better guide their decisions. However, the study has limitations, including its inability to account for variations in seed quality, labor skills, soil health metrics, irrigation water quality, and land degradation. Future research should focus on linkages between rural economies, extension approaches, and social networks, examine environmental consequences, soil properties, and productivity improvements resulting from DSR, and analyze yield and cost implications.

## Supporting information

**S1 Table. Impact of DSR adoption in Uttar Pradesh.** Effect of adopting DSR practices on income, production, and expenses associated with paddy cultivation in Uttar Pradesh. (DOCX)

**S2 Table. Impact of DSR adoption in Madhya Pradesh.** Effect of adopting DSR practices on income, production, and expenses associated with paddy cultivation in Madhya Pradesh. (DOCX)

**S3 Table. Impact of DSR adoption in Andhra Pradesh.** Effect of adopting DSR practices on income, production, and expenses associated with paddy cultivation in Andhra Pradesh. (DOCX)

**S4 Table. Impact of DSR adoption in Telangana.** Effect of adopting DSR practices on income, production, and expenses associated with paddy cultivation in Telangana. (DOCX)

## Acknowledgments

We acknowledge the support of our field team for collecting primary data to support this study.

## Author contributions

**Conceptualization:** Shiladitya Dey, Kumar Abbhishek, Suman Saraswathibatla.

**Data curation:** Shiladitya Dey.

**Formal analysis:** Shiladitya Dey, Debabrata Das.

**Funding acquisition:** Suman Saraswathibatla.

**Methodology:** Shiladitya Dey.

**Project administration:** Suman Saraswathibatla.

**Software:** Shiladitya Dey.

**Supervision:** Shiladitya Dey, Kumar Abbhishek, Suman Saraswathibatla, Debabrata Das.

**Validation:** Shiladitya Dey.

**Writing – original draft:** Shiladitya Dey, Kumar Abbhishek, Debabrata Das.

**Writing – review & editing:** Shiladitya Dey, Kumar Abbhishek, Debabrata Das.

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
