## [Decision Letter · Decision Letter 0]

4 Dec 2024

PONE-D-24-42084Economic Suitability of Direct Seeded Rice across Different Geographies in IndiaPLOS ONE

Dear Dr. Abbhishek,

Thank you for submitting your manuscript to PLOS ONE. After careful consideration, we feel that it has merit but does not fully meet PLOS ONE’s publication criteria as it currently stands. Therefore, we invite you to submit a revised version of the manuscript that addresses the points raised during the review process.

We look forward to receiving your revised manuscript.

Kind regards,

Jaipal Singh Choudhary, Ph.D.

Academic Editor

PLOS ONE

Journal Requirements:

2. You indicated that ethical approval was not necessary for your study. We understand that the framework for ethical oversight requirements for studies of this type may differ depending on the setting and we would appreciate some further clarification regarding your research. Could you please provide further details on why your study is exempt from the need for approval and confirmation from your institutional review board or research ethics committee (e.g., in the form of a letter or email correspondence) that ethics review was not necessary for this study? Please include a copy of the correspondence as an """"Other"""" file.

4. Please include captions for your Supporting Information files at the end of your manuscript, and update any in-text citations to match accordingly. Please see our Supporting Information guidelines for more information: http://journals.plos.org/plosone/s/supporting-information .

Additional Editor Comments:

Both reviewers have major concerns specially some overstated results and discussion need to be verified and re-written

Reviewers' comments:

Reviewer's Responses to Questions

**Comments to the Author**

1. Is the manuscript technically sound, and do the data support the conclusions?

Reviewer #1: Yes

Reviewer #2: Partly

2. Has the statistical analysis been performed appropriately and rigorously? 

Reviewer #1: Yes

Reviewer #2: Yes

3. Have the authors made all data underlying the findings in their manuscript fully available?

Reviewer #1: Yes

Reviewer #2: Yes

4. Is the manuscript presented in an intelligible fashion and written in standard English?

Reviewer #1: Yes

Reviewer #2: No

5. Review Comments to the Author

Reviewer #1: The manuscript deals with direct seeded rice feasibility as a sustainable substitute for puddled transplanted rice in India’s paddy production. The findings are promising for farmers and policymakers aiming to address labor and water scarcity in agriculture while promoting eco-friendly practices. The researchers collected extensively data from 537 households cultivating rice through DSR and PTR. They followed PSM and ESR methodology for analysis for proper outcome of their study, however, I found some limitations in their study which needs to be improved before acceptance.

1.L47 and 48: Rewrite sentence

2.Table 2: Yield 9kg/acre) in case of UP is exactly same when farmers sown rice through DSR and PTR, what are the factors that contributed to equal yield (when sample size is quite good), while weed and pest problem occurrence were more in DSR? Please incorporate explanation in discussion section for better understanding of readers.

3.In discussion section, it is written that farmers of UP are mostly having small land holding, and still they got equal yield, yield is one of the major factors in choosing new technology, then why farmers are still hesitating in adopting new technology, any reasons, incorporate it?

4.L468: The yield of rice is adversely affected by seed treatment; isn’t statement is contradictory?

5.While the study mentions economic and agronomic outcomes, it does not extensively explore how socioeconomic factors, such as education, access to technology, and extension services, impact dDSR adoption and success?

6.Compare data from farmers field and institute research trials located in these states across agro-climatic zones for better comparison, also targeted trials to identify specific adaptations needed for dDSR success? Tailoring techniques based on regional characteristics, such as soil type, climate, and water availability, could improve its suitability and encourage broader adoption.

7.Consider additional data variables such as soil health metrics, irrigation water quality, farmer education levels, and labor skills. Including these factors in analyses would provide a more comprehensive understanding of the variables influencing dDSR success.

Reviewer #2: The article needs through revision before considerations in the journal. I have gone through the ms and my commens is appended in the manuscript pdf. In my view, whole ms need to be re-write with suitable logic and data support. Most of places very old rreferences were given. In my view, delete all old references and cite relevant one. Avoild to use lumsum references and summary and conclusion should be presented in bullts form to better impact for international readers.

6. PLOS authors have the option to publish the peer review history of their article (what does this mean? ). If published, this will include your full peer review and any attached files.

**Do you want your identity to be public for this peer review?** For information about this choice, including consent withdrawal, please see our Privacy Policy .

Reviewer #1: **Yes: ** B. Lal

Reviewer #2: **Yes: ** Rakesh Kumar

---

## [Author Response · Author response to Decision Letter 1]

8 Jan 2025

Author’s Response to The Editor’s and Reviewer’s Comments

Thank you very much for your time and efforts in reviewing the MS. We have addressed all the concerns. The point to point response to all of the comments of both reviewers are given below.

Reviewer #1:

Comment: The manuscript deals with direct seeded rice feasibility as a sustainable substitute for puddled transplanted rice in India’s paddy production. The findings are promising for farmers and policymakers aiming to address labor and water scarcity in agriculture while promoting eco-friendly practices. The researchers collected extensively data from 537 households cultivating rice through DSR and PTR. They followed PSM and ESR methodology for analysis for proper outcome of their study, however, I found some limitations in their study which needs to be improved before acceptance.

Comment:1. L47 and 48: Rewrite sentence

Authors’ response: Done, Thank You. We have rewritten the sentences. Please refer to line no.: 47 & 48.

Comment: 2.Table 2: Yield 9kg/acre) in case of UP is exactly same when farmers sown rice through DSR and PTR, what are the factors that contributed to equal yield (when sample size is quite good), while weed and pest problem occurrence were more in DSR? Please incorporate explanation in discussion section for better understanding of readers.

Authors’ response: Done, Thank You. In UP, the rice yield (kg/acre) in a recently adopted DSR (last 3 years) and PTR (Table 2) is tending to a significant difference. In the case of DSR, yield is 1793kg/acre while for PTR it is 1739kg/acre, indicating a better nutrient use efficiency in case of DSR. Furthermore, puddling in PTR can lead to soil compaction, which negatively affects root growth and water infiltration. On the other hand, DDSR's roots grow deeper and stronger as they are not confined to the puddled soil. This can lead to better nutrient uptake and overall plant health, as well as increased yield. Also, inherent wetting and drying cycles in absence of ponded water in DSR promotes better microbial growth in the soil. The symbiotic relationship between microbes and plant roots leads to better nutrient availability for the plant, leading to better plant growth and yield than PTR. We have included these in the discussions section. (Lines: 280-287).

Yes, we agree that the weed and pest manifestation is high in DSR. The problem worsens slowly with multiple cycles of DSR, where weed-seed-bank in deeper soil gets activated. However, these are less prominent in initial years and can be controlled by following a proper weed management schedule. Our study observed that farmers in UP are well-trained in managing weeds in DSR plots. They apply pre and post-emergence weedicides to manage the weed growth, which is prevalent in the first 30 days after sowing.

Comment:3. In discussion section, it is written that farmers of UP are mostly having small land holding, and still they got equal yield, yield is one of the major factors in choosing new technology, then why farmers are still hesitating in adopting new technology, any reasons, incorporate it?

Authors’ response: Done. We agree that the farmers of UP are mostly having small land holding and yield is one of the major factors in choosing new technology. However, we observed that the delay in monsoon has devastating impacts in sandy loam and sandy clay loam soils of UP. There is low seed germination and farmers need to invest more in seed procurement and require additional labor for gap-filling. Nevertheless, intense rainfall after few weeks of sowing damages the re-sown seed. Additionally, socio-psychological factors, like the risk of losing yield if they fail to control weeds, risk of social exclusion from the community if they adopt a new practice, and difficulty in machine harvesting as DSR plots mature 10-15 days earlier than PTR. These are among few important factors responsible for slow adoption of DSR over PTR in UP. We have mentioned these in the revised MS. Please refer to lines: 446-453.

Comment: 4. L468: The yield of rice is adversely affected by seed treatment; isn’t statement is contradictory?

Authors’ response: Done. Thank you for your suggestion. We have rewritten the sentence. Please refer to lines: 492-495.

Comment: 5: While the study mentions economic and agronomic outcomes, it does not extensively explore how socioeconomic factors, such as education, access to technology, and extension services, impact dDSR adoption and success?

Authors’ response: Done. The impact of socio-economic factors on DSR adoption over PTR has been extensively covered in our previous publication (Dey et al., 2023). Therefore, in this study, our focus was to identify the economic suitability of DSR across four geographies. However, we have identified the impact of socio-economic constraints on DSR adoption for all four geographies in this study as well. Please refer to lines: 389-396, 400-409, 412-416.

Comment:6. Compare data from farmers field and institute research trials located in these states across agro-climatic zones for better comparison, also targeted trials to identify specific adaptations needed for dDSR success? Tailoring techniques based on regional characteristics, such as soil type, climate, and water availability, could improve its suitability and encourage broader adoption.

Authors’ response: Thank you for your suggestions. In fact, we are planning in the similar line. We are also doing some experimental trials considering the differences in the soil type, water, and climatic characteristics in all four locations. Once we complete the 2-year rigorous data analysis from the experimental plots, we will draft a manuscript on the same line. Nevertheless, we have mentioned it as a limitation of current study. Please refer to lines: 535-537.

Comment:7. Consider additional data variables such as soil health metrics, irrigation water quality, farmer education levels, and labor skills. Including these factors in analyses would provide a more comprehensive understanding of the variables influencing dDSR success.

Authors’ response: Thank you for your suggestions. We need to perform soil and water analysis in all locations to include these parameters. In the future, we will work on this and develop a separate manuscript. However, we added this as a limitation of our study. Please refer to lines: 533-535.

Reviewer #2

Comment: The article needs through revision before considerations in the journal. I have gone through the ms and my commens is appended in the manuscript pdf. In my view, whole ms need to be re-write with suitable logic and data support. Most of places very old rreferences were given. In my view, delete all old references and cite relevant one. Avoild to use lumsum references and summary and conclusion should be presented in bullts form to better impact for international readers.

Authors’ response: Thank you very much for your time and efforts in reviewing the MS. We have addressed all your concerns in revised MS Please, find point-to-point responses below and marked blue in revised MS draft.

Comment: What happened about Punjab, Haryana and eastern states of India

Authors’ response: Thank you for this suggestion. We concur that it would be advantageous to compare all rice-growing agro-climatic zones in a study. However, this study aimed to identify the economic suitability of DSR across four principal agro-climatic zones, each characterised by distinct soil types. These are our primary impact regions where we collaborate with farmers and various government departments. Moreover, the agricultural practices in these regions exhibit distinct variations. The government incentive for DSR cultivation in Punjab offers it a competitive advantage over traditional PTR methods. The land area in Punjab and Haryana is significantly larger than in these regions. In contrast, the meteorological and precipitation patterns in eastern India markedly diverge from those in these regions. Therefore, including Punjab, Haryana, and Eastern states in this analysis would not have yielded a comparable level playing field. Nonetheless, we acknowledge this as a weakness of the study and want to prepare a separate publication addressing it.

Comment: Puddle transplanting rice (PTR)

Authors’ response: Done. Thank You. Please refer to lines: 55-56.

Comment: The

Authors’ response: Done. Thank You. Please refer to line: 61.

Comment: pl cite recent references instead of very old

Authors’ response: Done. We have cited recent references in Table 1. (Line 73)

Comment: Pl write in bulleted form for better readability to the international readers

Authors’ response: Done, Thank you for your suggestion. We have added bulleted highlights as summary of the MS. Please find it in the Lines: 11-19.

Comment: Use recent references and delete old an irrelevant one

Authors’ response: Done. Recent references are included and the older ones have been deleted across the revised MS. For example, please refer to lines nos:575-588.

---

## [Editor Report · Decision Letter 1]

17 Jan 2025

PONE-D-24-42084R1Economic Suitability of Direct Seeded Rice across Different Geographies in IndiaPLOS ONE

Dear Dr. Abbhishek,

Thank you for submitting your manuscript to PLOS ONE. After careful consideration, we feel that it has merit but does not fully meet PLOS ONE’s publication criteria as it currently stands. Therefore, we invite you to submit a revised version of the manuscript that addresses the points raised during the review process.

We look forward to receiving your revised manuscript.

Kind regards,

Jaipal Singh Choudhary, Ph.D.

Academic Editor

PLOS ONE

Journal Requirements:

Additional Editor Comments:

Thank you for revise of manuscript and substantial improvement but still it has limitation in in discussion and conclusion part.

Separate the result and discussion part in to different subheading so that result can be discussed well in in discussion part.

Conclusion is to be conclusive so it should be short with only research findings oriented and research way forwards.

---

## [Author Response · Author response to Decision Letter 2]

11 Feb 2025

Thank you very much for your time and efforts in reviewing the MS. We have addressed all the concerns. The point-to-point response to all of the comments of both reviewers is given below.

Comment: A rebuttal letter that responds to each point raised by the academic editor and reviewer(s). You should upload this letter as a separate file labeled 'Response to Reviewers'.

Authors’ response: Done. Thank you for the suggestion.

Comment: A marked-up copy of your manuscript that highlights changes made to the original version. You should upload this as a separate file labeled 'Revised Manuscript with Track Changes'.

Authors’ response: Done. Thank you for the suggestion.

Comment: An unmarked version of your revised paper without tracked changes. You should upload this as a separate file labeled 'Manuscript'.

Authors’ response: Done. Thank you for the suggestion.

Comment: Please review your reference list to ensure that it is complete and correct. If you have cited papers that have been retracted, please include the rationale for doing so in the manuscript text, or remove these references and replace them with relevant current references. Any changes to the reference list should be mentioned in the rebuttal letter that accompanies your revised manuscript. If you need to cite a retracted article, indicate the article’s retracted status in the References list and also include a citation and full reference for the retraction notice.

Authors’ response: Done. We have not cited any paper that has retracted

Comment: Thank you for revise of manuscript and substantial improvement but still it has limitation in in discussion and conclusion part.

Authors’ response: Done. Thank you for your suggestion. We have rewritten the discussion and conclusion part. Please refer to lines no.: 220-473

Comment: Separate the result and discussion part in to different subheading so that result can be discussed well in in discussion part.

Authors’ response: Done. We have separated the results and discussion under different sub-headings. Please refer to line no.: 220-253, 267-297, 300-316, 322-450.

Conclusion is to be conclusive so it should be short with only research findings oriented and research way forwards.

Authors’ response: Done. We have rewritten the conclusion. Please refer to lines no.: 452-473.

---

## [Editor Report · Decision Letter 2]

16 Feb 2025

PONE-D-24-42084R2Economic Suitability of Direct Seeded Rice across Different Geographies in IndiaPLOS ONE

Dear Dr. Abbhishek, 

Thank you for submitting your manuscript to PLOS ONE. After careful consideration, we feel that it has merit but does not fully meet PLOS ONE’s publication criteria as it currently stands. Therefore, we invite you to submit a revised version of the manuscript that addresses the points raised during the review process.

We look forward to receiving your revised manuscript.

Kind regards,

Jaipal Singh Choudhary, Ph.D.

Academic Editor

PLOS ONE

Journal Requirements:

Additional Editor Comments:

I unable to see what changes authors have been done in revised manuscript. There were clear cut instruction given that "You should upload this as a separate file labeled 'Revised Manuscript with Track Changes'" and again authors submitted manuscript with all blue highlighted manuscript. Considering the importance of time of reviewer and editors this is the last opportunity to authors for revision of manuscript as suggested.

---

## [Author Response · Author response to Decision Letter 3]

28 Feb 2025

Dear Editor and Reviewers,

We highly appreciate your valuable time and efforts put in the revision of this MS. We believe that your suggestions have made the MS more focused and reader-apt. We have made the revision as per your suggestions

Comment: A rebuttal letter that responds to each point raised by the academic editor and reviewer(s). You should upload this letter as a separate file labeled 'Response to Reviewers'.

Authors’ response: Done. Thank you for the suggestion.

Comment: A marked-up copy of your manuscript that highlights changes made to the original version. You should upload this as a separate file labeled 'Revised Manuscript with Track Changes'.

Authors’ response: Done. Thank you for the suggestion.

Comment: An unmarked version of your revised paper without tracked changes. You should upload this as a separate file labeled 'Manuscript'.

Authors’ response: Done. Thank you for the suggestion.

Journal Requirements:

Comment: Please review your reference list to ensure that it is complete and correct. If you have cited papers that have been retracted, please include the rationale for doing so in the manuscript text, or remove these references and replace them with relevant current references. Any changes to the reference list should be mentioned in the rebuttal letter that accompanies your revised manuscript. If you need to cite a retracted article, indicate the article’s retracted status in the References list and also include a citation and full reference for the retraction notice.

Authors’ response: Done. We have crosschecked all the references and additionally deleted those not cited in the text from the references list.

Additional Editor Comments:

Comment: I unable to see what changes authors have been done in revised manuscript. There were clear cut instruction given that "You should upload this as a separate file labeled 'Revised Manuscript with Track Changes'" and again authors submitted manuscript with all blue highlighted manuscript. Considering the importance of time of reviewer and editors this is the last opportunity to authors for revision of manuscript as suggested.

Authors’ response: Done. Thank you for your suggestions. This time, we have uploaded 'Revised Manuscript with Track Changes' and also the unmarked version of the revised paper without tracked changes.

Also, we would like to inform you that based on the previous comments, we have separated the results and discussion under different sub-headings. Please refer to line no.: 254-288, 291-305, 308-313, 335-338, 341-342, 345-353, 357-389, 404-405, 427-453, 459-674. Also, as per the suggestions, we have rewritten the conclusion. Please refer to lines no.: 717-742. Please note that all the above-mentioned lines are as per the revised manuscript with track changes.

---

## [Editor Report · Decision Letter 3]

7 Mar 2025

Economic Suitability of Direct Seeded Rice across Different Geographies in India

PONE-D-24-42084R3

Dear Dr. Kumar,

We’re pleased to inform you that your manuscript has been judged scientifically suitable for publication and will be formally accepted for publication once it meets all outstanding technical requirements.

Kind regards,

Jaipal Singh Choudhary, Ph.D.

Academic Editor

PLOS ONE
---

## [Editor Report · Acceptance letter]

PONE-D-24-42084R3

PLOS ONE

Dear Dr. Abbhishek,

I'm pleased to inform you that your manuscript has been deemed suitable for publication in PLOS ONE. Congratulations! Your manuscript is now being handed over to our production team.

Kind regards,

on behalf of

Dr. Jaipal Singh Choudhary

Academic Editor

PLOS ONE